# Remote Estimation of Biomass in Winter Oilseed Rape (*Brassica napus* L.) Using Canopy Hyperspectral Data at Different Growth Stages

**Yi Ma, Shenghui Fang \*, Yi Peng, Yan Gong and Dong Wang**

School of Remote Sensing and Information Engineering, Wuhan University, Wuhan 430079, China; mayi@whu.edu.cn (Y.M.); ypeng@whu.edu.cn (Y.P.); gongyan@whu.edu.cn (Y.G.); timdong@whu.edu.cn (D.W.)
\* Correspondence: shfang@whu.edu.cn; Tel.: +86-138-0719-0453

**Abstract:** The dry aboveground biomass (AGB) is an important parameter in assessing crop growth and predicting yield. This study aims to ascertain the optimal methods for the spectroscopic estimation of winter oilseed rape (WOR) biomass. The different fertilizer-N gradients WOR were planted to collect biomass data and canopy hyperspectral data in two years of field experiments. Correlation analyses and partial least squares regression (PLSR) were performed between canopy hyperspectral data and AGB, and the linear and non-linear regression models simulated the quantitative relation between the vegetation indices (VIs) and AGB at four different growth stages (seeding, bolting, flowering, and pod stage). The results indicated that VIs that were derived from canopy hyperspectral data could estimate AGB accurately: (1) At the seeding and bolting stage, the CIred edge showed excellent performance with the higher accuracy ($R^2$ ranged from 0.60–0.95) as compared to the other six VIs (Green chlorophyll index (CIgreen), normalized difference vegetation index (NDVI), Green normalized difference vegetation index (GNDVI), ratio vegetation index (RVI), DVI, and soil adjusted vegetation index (SAVI)); (2) Correlation analyses and PLSR can effectively extract the feature wavelengths (800 nm and 1200 nm) for biomass estimation. The modified vegetation indices NDVI (800, 1200) significantly improved AGB estimation accuracy ($R^2 > 0.80$, RMSE < 1530 kg/hm², RPD > 2.3) without saturation phenomenon at the total for four stages, and retained good robustness and reduced the influence of flower and pod for estimating AGB; (3) it was vital to pay more attention to the near-infrared (NIR) bands that could represent WOR growth phenology, and selecting suitable VIs and modeling algorithms could also have a relatively large effect on the success of AGB estimation. The overall results indicated that WOR AGB could be reliably estimated by canopy hyperspectral data, although the plant architecture and coverage of WOR were significantly different during its entire growing period.

**Keywords:** winter oilseed rape; aboveground biomass; canopy hyperspectral data; correlation analysis; partial least squares regression

## 1. Introduction

Winter oilseed rape (WOR) is one of the major commercial crops, being grown mainly in temperate regions [1–3]. It is cultivated mostly for its oil-rich seeds that are widely used for food, biofuel, and medicine [4,5]. The dry aboveground biomass (AGB) is an important parameter in indicating crop growth status, farmers need crop biomass information at different growth stages for guiding their applying fertilizer, and the early estimation of AGB can also be utilized for yield prediction [6]. Traditionally, AGB estimation was based on ground destructive sampling, which was both time-consuming and unsuitable for large areas [7–9]. In contrast, with the development

of remote sensing technology, the high resolution, strong continuity, and massive remote sensing information provided a cost-effective method for avoiding sampling bias and opened a new perspective for quantifying and estimating AGB at both the local and regional scale [10]. Remote sensing had been proven to be a reliable method in monitoring plant growth and nutritional status over large areas [11], and is gradually used to assess plant and leaf nitrogen concentration, chlorophyll density for fertilization, and irrigation of WOR [12–14].

Most previous studies adopted large-scale satellite image for WOR AGB estimation. Zhang et al. [15] extracted 27 parameters from the C band synthetic aperture radar (SAR) image to estimate AGB with a determination coefficient ($R^2$) of 0.765 and root mean squared error (RMSE) of 73.20 $g/m^2$. The fully polarimetric SAR data was also used to estimate dry and fresh biomass with an $R^2$ of 0.85 and 0.76 and RMSE of 41.6 $g/m^2$ and 533.5 $g/m^2$, respectively [16]. However, the use of the SAR image remained challenging due to its high costs and algorithm complexity for large scale biomass estimation; it was also hypersensitive to the underlying surface, which might greatly influence the accuracy of AGB estimation [17]. When compared with SAR satellite sensors, high spatial resolution optical satellite sensors had a relatively shorter revisiting cycle and they were efficient in capturing finer plant growth information [18]. Han et al. [19] collected the optimum vegetation indices (VIs) from the time-series high spatial resolution satellite image (the Pleiades-1, Worldview-2, and Spot-6) for the whole growth period of WOR AGB estimation, with the accuracy of inversion model ($R^2$ = 0.77 and RMSE = 104.64 $g/m^2$). Although these studies had demonstrated the successful estimation of biomass by satellite image, the satellite image was still difficult and expensive to acquire crop growth information of small plots due to a long visiting cycle and cloud coverage, especially for the detailed canopy structural information [20,21]. The plant architecture and coverage of WOR were significantly different during its entire growing period since leaves, buds, flowers, and pods appeared gradually [22]. When considering these restrictions, the remote sensing techniques that are based on canopy hyperspectral data are needed for accurately monitoring WOR AGB.

In general, canopy hyperspectral data was commonly used in precision agriculture to non-destructively estimate agronomic parameters, including AGB [23,24]. The results of several studies showed that hyperspectral VIs were considered to be more sensitive in quantifying vegetation biomass [25,26]. For example, the normalized difference vegetation index (NDVI) that was associated with the visible and near-infrared wavebands was mostly recommended in monitoring the biophysical characteristics (biomass, leaf area, leaf gap fraction, and plant cover) [27]. Cho et al. [28] found that a linear model based on NDVI (740 nm, 771 nm) produced higher accuracy than that on a traditional NDVI (665 nm, 801 nm) for grass biomass estimation. Fu et al. [6] also showed that a linear model based on NDVI (1097 nm, 980 nm) had a better predictive performance in the estimation of winter wheat biomass ($R^2$ = 0.760, RMSE = 0.279 $kg/m^2$). However, the major problem of using the NDVI was that it became saturated for vegetation with moderate to high density [29]. Gitelson et al. [30] and Viña A et al. [31] developed the red edge chlorophyll index (CIred edge) and green chlorophyll index (CIgreen) to overcome the saturation phenomenon, which could estimate the wide range of biomass and chlorophyll of green leaf accurately. Besides, soil adjusted vegetation index (SAVI), including a background adjustment factor L, was calculated to reduce the influence of soil background. Kumar et al. [32] estimated the plantation biomass that is based on the SAVI and ratio vegetation index (RVI) by regression analysis ($R^2$ = 0.813). It was difficult to extract features from multidimensional hyperspectral data. Wold [33] proposed a refined methodology (Partial Least Squares Regression, PLSR) that could obtain important signals and develop reliable regression models, various studies have demonstrated that PLSR was a powerful statistical technique in extracting sensitive wavelengths and reducing the influence of external environmental disturbance for a higher estimation accuracy of plant biophysical characteristics [17,34–36]. WOR is very different from rice, wheat, and soybean, as it is a broadleaf plant with an obvious distinctive change of canopy structure during its entire growth period [37], and yellow flowers and green pods seriously affect the canopy hyperspectral data of WOR [38,39]. To date, few studies take into account the growth stages for spectroscopic WOR AGB

estimation, and it is unclear which method can effectively extract the feature wavelengths in reducing the influence of flower and pod.

Thus, the substantial objectives of this study were to (1) evaluate the ability of commonly used hyperspectral VIs for WOR AGB estimation at different stages, and prove whether VIs could overcome the saturation phenomenon; (2) analyze the effects of flowers and pods on canopy hyperspectral data, extract the feature wavelengths by correlation analysis and PLSR, and then construct new VIs for WOR AGB estimation accurately.

## 2. Materials and Methods

### 2.1. Experiment Design

Two experiments were conducted in this study. The experiment one was conducted in Wuxue city (30.1059° N, 115.5864° E), Hubei province, central China (Figure 1b) from September 2014 to May 2015. The 24 individual plot area was 20 m$^2$ (10.0 m × 2.0 m), which were planted with eight gradients of fertilizer-N, with each N rate being repeated in three distributed plots. Canopy hyperspectral data and AGB data were measured on December 8 in 2014, January 15, March 12, and April 14 in 2015. These dates corresponded to the four growth stages: seeding stage (eight leaf stage and 10 leaf stage), flowering stage, and pod stage.

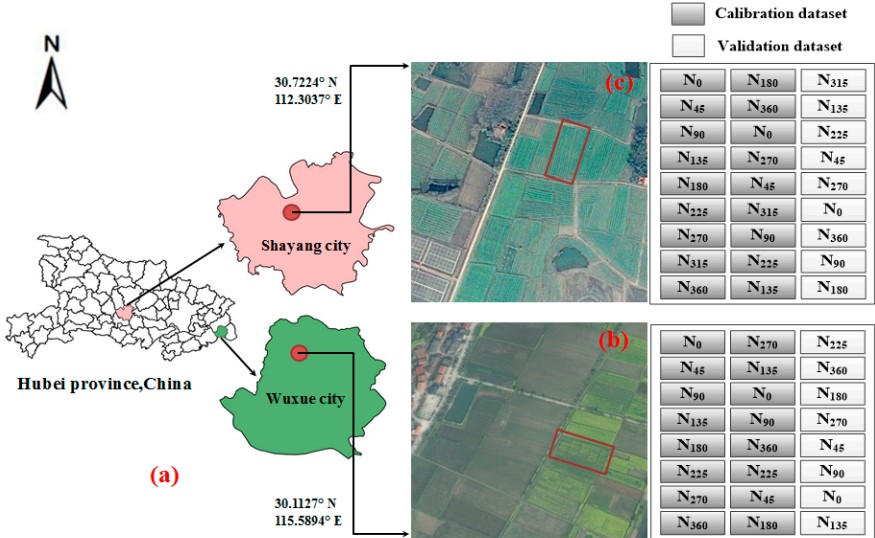

**Figure 1.** Experimental design: (**a**) Experiment location map; (**b**) Experiment one; and, (**c**) Experiment two.

The experiment two was conducted in Shayang city (30.7224° N, 112.3037° E), Hubei province, central China ((Figure 1c) from September 2015 to May 2016. The 27 individual plot area was 30 m$^2$ (15.0 m × 2.0 m), which were planted with nine gradients of fertilizer-N, with each N rate being repeated in three distributed plots. Canopy hyperspectral data and AGB data were measured on November 26 and December 14 in 2015, January 7, January 24, February 18, March 18, and April 18 in 2016. These dates corresponded to seven growth stages: seeding stage (seven leaf stage, eight leaf stage, 10 leaf stage, 11 leaf stage), bolting stage, flowering stage, and pod stage.

A widely used WOR variety "Huayouza No. 9" was transplanted into the tilled field with a unified density of 112,500 plants hm$^{-2}$ for the two growing seasons. The specific N treatments were shown in Table 1, for all treatments, monocalcium phosphate and potassium chloride were applied at 90 kg P$_2$O$_5$ hm$^{-2}$, 120 kg K$_2$O hm$^{-2}$, and 1.6 kg B hm$^{-2}$. Other management decisions (pest control and herbicide application) followed the local standard practices.

**Table 1.** Details of two field experiments.

| Year and Site | Plots | Fertilizer-N level (kg/hm$^2$) | The Major Chemical Properties of Soil Layer (0–20 cm) |
|---|---|---|---|
| 2014–2015 Wuxue city | The 24 individual plot area: 20 m$^2$ (10.0 m × 2.0 m) | 8 fertilizer-N levels: 0, 45, 90, 135, 180, 225, 270, 360 | pH:5.40 Organic matter: 30.08 g/kg Total N: 1.72 g/kg Olsen-P: 13.80 mg/kg NH4OAc-K: 92.10 mg/kg |
| 2015–2016 Shayang city | The 27 individual plot area: 30 m$^2$ (15.0 m × 2.0 m) | 9 fertilizer-N levels: 0, 45, 90, 135, 180, 225, 270, 315, 360 | pH:5.88 Organic matter: 18.02 g/kg Total N: 0.96 g/kg Olsen-P: 11.79 mg/kg NH4OAc-K: 85.97 mg/kg |

## 2.2. Experiment Data

### 2.2.1. Canopy Hyperspectral Data Acquisition

The hyper-spectrometer ASD FieldSpec 4 (ASD Inc., Boulder, Colorado, USA) was used to collect the canopy spectral data, with the spectrum ranging from 350–2500 nm. The measurements were conducted on sunny days from 10:00 am to 2:00 pm. A white reference panel (Chemical composition: $BaSO_4$, Size: 25.4 × 25.4 cm) was used to make relative radiometric correction before each measurement. The hyper-spectrometer was positioned above the WOR canopy at a height of 1 m with a 25° field-of-view of 0.1544 m$^2$ area. Five spectral measurements were averaged to reduce the uncertainty error of spectral measurement in every plot. In this study, the spectral waveband was set in the range of visible light and near-infrared shortwaves (400–1300 nm), with a high signal to noise ratio.

### 2.2.2. Dry Aboveground Biomass (AGB) Data

After taking ASD spectral measurements, four sampled plants from each plot were cut at ground level and then sent to laboratory immediately. These samples were dried for 30 min at 105 °C to deactivate enzymes and then dried at 70 °C until its weight did not change. Their dry biomass was subsequently weighed. Aboveground biomass (kg/hm$^2$) was determined by dividing the dry biomass weight (kg) by the unified density (112,500 plants per hm$^{-2}$).

### 2.2.3. Calibration and Validation Dataset

In the study, 24 plots (eight N treatments, three replications) and 27 plots (nine N treatments, three replications) were sampled in 2014–2015 and 2015–2016 growing seasons, respectively. The plots were divided into a 2:1 ratio as the calibration dataset (the first two replications) and the validation dataset (the third replication), the calibration and validation dataset are shown in Figure 1b,c and Table 2.

**Table 2.** Number of calibration and validation dataset at different growth stages.

| Growth Stages | Number of Calibration Dataset | Number of Validation Dataset | Study Site | Sampling Dates (month/day/year) | Number of Samples |
|---|---|---|---|---|---|
| Seeding stage (n = 156) | n = 104 | n = 52 | Wuxue city | 12/8/2014 | 24 |
| | | | Wuxue city | 1/15/2015 | 24 |
| | | | Shayang city | 11/26/2015 | 27 |
| | | | Shayang city | 12/14/2015 | 27 |
| | | | Shayang city | 1/7/2016 | 27 |
| | | | Shayang city | 1/24/2016 | 27 |
| Bolting stage (n = 27) | n = 18 | n = 9 | Shayang city | 2/18/2016 | 27 |
| Flowering stage (n = 51) | n = 34 | n = 17 | Wuxue city | 3/12/2015 | 24 |
| | | | Shayang city | 3/18/2016 | 27 |
| Pod stage (n = 51) | n = 34 | n = 17 | Wuxue city | 4/14/2015 | 24 |
| | | | Shayang city | 4/18/2016 | 27 |

*2.3. Research Methods*

2.3.1. Vegetation Indices (VIs)

Vegetation indices refers to the combination of certain waveband reflectance, which is related to the pigment of plant leaves, photosynthesis, and plant nutrition status [40]. In this study, the seven commonly used vegetation indices were selected for monitoring AGB, which have clear physical significance and higher recognition. The formulas and references of the vegetation indices are listed in Table 3.

**Table 3.** Vegetation indices (VIs) list.

| Vegetation Indices | Formulas | References |
|---|---|---|
| Red edge chlorophyll index (CIred edge) | $(R800/R725) - 1$ | [30] |
| Green chlorophyll index (CIgreen) | $(R800/R550) - 1$ | [41] |
| Normalized Difference Vegetation Index (NDVI) | $(R800 - R670)/(R800 + R670)$ | [42] |
| Green normalized difference vegetation index (GNDVI) | $(R780 - R550)/(R780 + R550)$ | [43] |
| Ratio vegetation index (RVI) | $R750/R550$ | [44] |
| Difference Vegetation Index (DVI) | $R800 - R670$ | [45] |
| Soil-Adjusted Vegetation Index (SAVI) | $(1 + L)(R890 - R670)/(R890 + R670 + L), L = 0.5$ | [46] |

Note: "R" represents the wavelength corresponding spectral reflectance values.

2.3.2. The Process of Modeling and Assessment

Correlation analyses were performed between canopy hyperspectral data and AGB, and the linear and non-linear (logarithmic, parabolic, power, and exponential) regression statistical models simulated the quantitative relation between VIs and AGB at different growth stages.

Linear model:

$$y = ax + b \tag{1}$$

Logarithmic model:

$$y = a \times \log(x) + b \tag{2}$$

Parabolic model:

$$y = ax^2 + bx + c \tag{3}$$

Power model:

$$y = ax^b \tag{4}$$

Exponential model:

$$y = ae^x \tag{5}$$

Note: "y" represents the AGB values, "x" represents the VIs values, "a, b, c" means the model fitting parameters, and "e" represents the natural index value

The performances of the various regression models were evaluated by the coefficient of determination ($R^2$), root mean square error (RMSE) for calibration (RMSEC), validation (RMSEV), and the ratio of prediction to deviation (RPD). Mathematically, the higher $R^2$ and RPD correspond to the smaller RMSE, and thus represent better model accuracy [47].

Data modeling process: firstly, the linear and non-linear regression models were established based on VIs and AGB of the calibration dataset, we selected the best model forms (linear, logarithmic, parabolic, power, and exponential) by comparing the $R^2$ and RMSEC, and then acquired the corresponding calibration models parameters. Secondly, the predicted AGB value was calculated by the VIs of the validation dataset with the application of the selected best calibration models forms and its parameters. Thirdly, the linear regression models were established based on the predicted AGB value and the original AGB of validation dataset, and the $R^2$, RMSEV, and RPD were used to evaluate the accuracy of validation models.

The following equations were used to calculate $R^2$, RMSE, and RPD:

$$R^2 = 1 - \frac{\sum_{i=1}^{n} (y_i - x_i)^2}{\sum_{i=1}^{n} (y_i - \overline{y})^2} \tag{6}$$

$$RMSE = \sqrt{\frac{\sum_{i=1}^{n} (y_i - x_i)^2}{n}} \tag{7}$$

$$RPD = \frac{STD}{RMSEV} \tag{8}$$

Where $x_i$ and $y_i$ are the estimated and measured AGB values, respectively, $\overline{y}$ and STD is the average and standard deviation of the measured AGB values, RMSEV represents the root mean square error of validation model, and n is the sample number, respectively.

The linear and non-linear regression was performed using SPSS 19.0 software (SPSS, Chicago, IL, USA). Graphics were prepared using the Origin 9.1 software program (Origin 9.1, OriginLab Corporation, Northampton, USA).

## 3. Results

### 3.1. Statistical Values of AGB

Along the development of WOR phenology, Table 4 showed that the mean of AGB increased from 1444.00 to 8700.45 kg/hm$^2$, the standard deviation of AGB also increased from 1026.08 to 4480.54 kg/hm$^2$, and the Coef. of Variation also exhibited wide variation with a range of values from 71.05%–51.50%. The wide range of AGB statistical characteristics showed that the amount of nitrogen fertilizer significantly affected AGB, and the relatively discretized statistical characteristics may influence the accuracy of AGB estimation at different growth stages.

**Table 4.** The statistical characteristics of aboveground biomass (AGB) from 2014 to 2016.

| Growth Stages | Minimum (kg/hm$^2$) | Maximum (kg/hm$^2$) | Mean (kg/hm$^2$) | Standard Deviation | Variance | Coef. of Variation |
|---|---|---|---|---|---|---|
| Seeding stage (n = 156) | 179.00 | 3982.00 | 1444.00 | 1026.08 | 1052847.38 | 71.05 |
| Bolting stage (n = 27) | 527.00 | 5428.00 | 3052.85 | 1798.74 | 3235464.82 | 58.92 |
| Flowering stage (n = 51) | 674.00 | 10286.00 | 4589.64 | 2743.17 | 7524995.81 | 59.77 |
| Pod stage (n = 51) | 1445.63 | 17402.00 | 8700.45 | 4480.54 | 20075254.99 | 51.50 |

### 3.2. Representative Spectral Reflectance

As Figure 2 displays, the canopy spectral reflectance of WOR had obvious differences at different growth stages, the canopy spectral reflectance was mainly affected by leaf chlorophyll and growth status [48]. At seeding, bolting, and pod stages, the canopy reflectance presented the spectral curves of typical green vegetation. At the flower stage, the canopy reflectance was greatly affected by flower, the reflectance increased significantly at visible wavelengths (500–730 nm), while it decreased at NIR wavelengths (800–1300 nm). Generally, the canopy spectral reflectance sensibly responded to the wide variation of WOR growth status and gradually got higher in the visible region and lower in the NIR region with the process of the growth period.

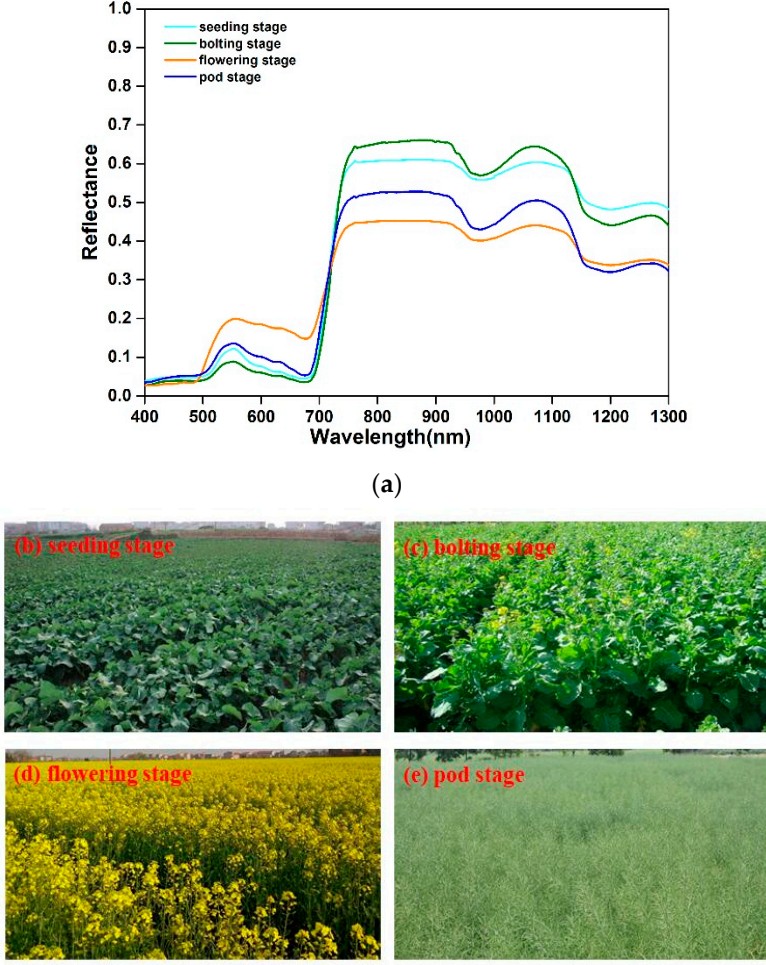

**Figure 2.** Canopy spectral reflectance and field pictures at four different stages. (**a**) Canopy spectral reflectance at seeding, bolting, flowering and pod stages; (**b**) Field picture at seeding stage; (**c**) Field picture at bolting stage; (**d**) Field picture at flowering stage; (**e**) Field picture at pod stage.

### 3.3. Correlation Analysis for Hyperspectral Waveband and AGB

As Figure 3 displays the correlation coefficient between AGB and canopy spectral reflectance at four different stages, at seeding stage, the correlation coefficient in the range of 400–706 nm showed a negative correlation, and the maximum negative correlation was about $-0.37$, being located at 661 nm. In the waveband range of 690 to 750 nm, the correlation coefficient changed rapidly from negative correlation to significant positive correlation, with the increase of wavelength. In the NIR plateau (750–1300 nm), the correlation coefficient was relatively stable and close to 0.6.

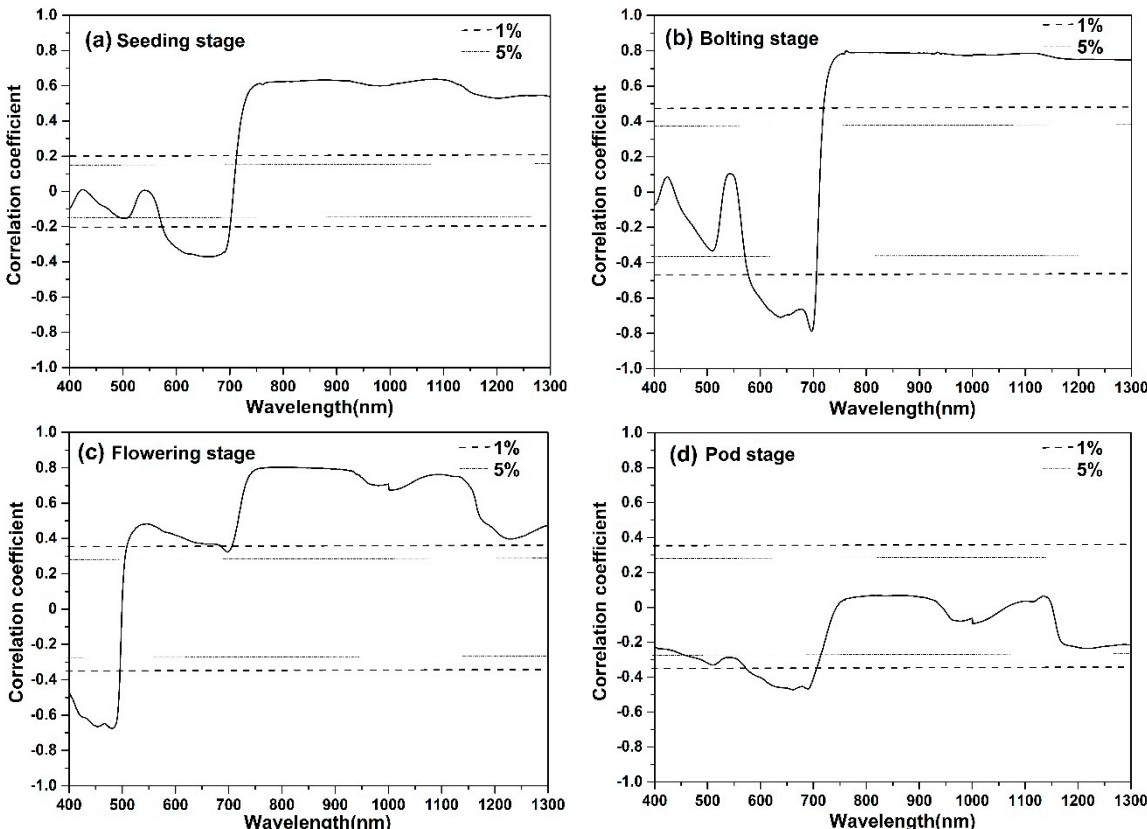

**Figure 3.** Correlation coefficient results of AGB and canopy spectral reflectance at four different stages: (**a**) Seeding stage; (**b**) Bolting stage; (**c**) Flowering stage; and, (**d**) Pod stage.

At bolting stage, the correlation coefficient analysis was analogous to that of the seeding stage; in the NIR plateau (750–1300 nm), the extremely significant positive correlation coefficient was also relatively stable, being close to 0.8.

At the flowering stage, the correlation coefficient changed rapidly from negative correlation to positive correlation with the increase of wavelength (480–551 nm). In the waveband range of 550 to 700 nm, the positive correlation coefficient slightly decreased, while the correlation coefficient increased rapidly with the increase of wavelength (710–750 nm). In the NIR plateau (750–1300 nm), the positive correlation coefficient was always higher than 0.4, but there were two big low near 970 nm and 1230 nm.

At the pod stage, the correlation coefficient in the range of 400–1300 nm was insignificant, except for in the range of 570–710 nm, and the negative correlation remained in the almost waveband range of 400 to 1300 nm.

The correlation coefficient analysis indicated that the relationship between NIR bands (750–1300 nm) and AGB was relatively stable at four different stages and it might optimize NIR bands combination to improve AGB estimation accuracy.

### 3.4. Relationship Between the Seven VIs and AGB at Four Different Stages

We used linear and nonlinear (logarithmic, parabolic, power, and exponential) regression analysis to investigate VIs with AGB at four different growth stages. Table 5 showed the best AGB estimation models, respectively. The results indicated that the accuracy of AGB estimation models that were based on the seven VIs were different at different growth stages.

**Table 5.** Regression modeling analysis between the seven VIs and AGB.

| Growth Period | Model Number | Vegetation Index | Model | $R_{cal}^2$ | RMSEC | $R_{val}^2$ | RMSEV | RPD |
|---|---|---|---|---|---|---|---|---|
| Seeding stage | 1 | CIred edge | $Y = 102.97e^{5.47X}$ | 0.75 | 516.02 | 0.60 | 667.45 | 1.49 |
| | 2 | CIgreen | $Y = 23.79X^{1.75}$ | 0.46 | 765.90 | 0.43 | 754.32 | 1.32 |
| | 3 | NDVI | $Y = 0.15e^{10.86X}$ | 0.44 | 782.97 | 0.35 | 808.95 | 1.23 |
| | 4 | GNDVI | $Y = 66244.38X^2 - 71070.06X + 19149.09$ | 0.70 | 567.58 | 0.62 | 634.92 | 1.57 |
| | 5 | RVI | $Y = 140.90X^2 - 532.25X + 468.94$ | 0.66 | 603.71 | 0.60 | 645.25 | 1.54 |
| | 6 | DVI | $Y = 1030.32X^2 + 3315.29X - 430.26$ | 0.38 | 820.85 | 0.48 | 719.00 | 1.38 |
| | 7 | SAVI | $Y = 9193.38X^2 - 6090.45X + 1306.71$ | 0.42 | 794.13 | 0.51 | 698.61 | 1.42 |
| Bolting stage | 8 | CIred edge | $Y = 9197.71X^{2.35}$ | 0.90 | 548.12 | 0.95 | 563.53 | 3.36 |
| | 9 | CIgreen | $Y = 28.60X^2 - 144.13X + 510.72$ | 0.60 | 1116.29 | 0.68 | 1200.59 | 1.58 |
| | 10 | NDVI | $Y = 0.01e^{14.69X}$ | 0.74 | 909.41 | 0.83 | 931.60 | 2.03 |
| | 11 | GNDVI | $Y = 0.002e^{19.10X}$ | 0.76 | 860.64 | 0.85 | 917.05 | 2.06 |
| | 12 | RVI | $Y = 414.29X^2 - 3126.00X + 6200.30$ | 0.75 | 880.71 | 0.84 | 935.99 | 2.02 |
| | 13 | DVI | $Y = -544.29X^2 + 10100.64X - 1593.67$ | 0.65 | 1045.51 | 0.64 | 1116.31 | 1.70 |
| | 14 | SAVI | $Y = 19512.53X^2 - 12498.74X + 2389.74$ | 0.68 | 1000.69 | 0.71 | 1039.45 | 1.82 |
| Flowering stage | 15 | CIred edge | $Y = 15030.54X^2 + 4155.57X + 617.51$ | 0.51 | 1930.70 | 0.73 | 1383.12 | 1.94 |
| | 16 | CIgreen | $Y = -40.19X^2 + 1635.22X - 5133.96$ | 0.46 | 2035.52 | 0.69 | 1506.24 | 1.78 |
| | 17 | NDVI | $Y = 38617.39X^2 - 32985.07X + 10536.23$ | 0.07 | 2670.14 | 0.22 | 2313.93 | 1.16 |
| | 18 | GNDVI | $Y = 80802.89X^2 - 62164.29X + 15637.80$ | 0.09 | 2647.08 | 0.21 | 2331.43 | 1.15 |
| | 19 | RVI | $Y = 665.18X^2 - 2035.16X + 5086.06$ | 0.06 | 2689.12 | 0.20 | 2359.91 | 1.14 |
| | 20 | DVI | $Y = -32022.02X^2 + 49340.35X - 8223.08$ | 0.61 | 1725.98 | 0.65 | 1562.56 | 1.72 |
| | 21 | SAVI | $Y = -82431.02X^2 + 104058.89X - 25617.79$ | 0.44 | 2080.93 | 0.54 | 1794.47 | 1.50 |
| Pod stage | 22 | CIred edge | $Y = 19903.34X^2 + 12952.22X + 30.53$ | 0.60 | 2885.85 | 0.57 | 2789.80 | 1.55 |
| | 23 | CIgreen | $Y = 369.40X^2 - 3109.09X + 9343.05$ | 0.63 | 2757.06 | 0.46 | 3128.60 | 1.39 |
| | 24 | NDVI | $Y = 0.15e^{13.40X}$ | 0.67 | 2627.01 | 0.52 | 2969.42 | 1.46 |
| | 25 | GNDVI | $Y = 300474.05X^2 - 255779.54X + 54925.75$ | 0.68 | 2577.58 | 0.61 | 2768.50 | 1.57 |
| | 26 | RVI | $Y = 9229.40X - 25666.00$ | 0.68 | 2570.50 | 0.61 | 2826.18 | 1.53 |
| | 27 | DVI | $Y = -263029.22X^2 + 210863.21X - 31888.76$ | 0.25 | 3927.09 | 0.20 | 3959.06 | 1.10 |
| | 28 | SAVI | $Y = -229623.73X^2 + 281044.12X - 76153.15$ | 0.22 | 4003.84 | 0.37 | 3498.71 | 1.24 |

At the seeding stage, the $R^2_{cal}$ and RMSEC of calibration models varied, respectively, from 0.38–0.75 and 516.02–820.85 kg/hm$^2$, while the $R^2_{val}$, RMSEP, and RPD of validation models were between 0.35–0.62, 634.92–808.95 kg/hm$^2$, and 1.23–1.57. Through comparing the accuracy of models, models that were based on CIred edge and Green normalized difference vegetation index (GNDVI) proved to have good fitting characteristics, relatively high precision models.

At the bolting stage, $R^2_{cal}$ and RMSEC varied, respectively, from 0.60–0.90 and 548.12–1116.29 kg/hm$^2$, while $R^2_{val}$, RMSEV, and RPD were between 0.64–0.95, 563.53–1200.59 kg/hm$^2$, and 1.58–3.36; model 8 based on CIred edge proved to have good fitting characteristics, relatively high precision model.

At the flowering stage, the accuracy of AGB estimation models that were based on VIs were relatively low, and $R^2_{cal}$ and RMSEC varied, respectively, from 0.06–0.51 and 1725.98–2689.12 kg/hm$^2$, while $R^2_{val}$, RMSEV, and RPD of validation models were between 0.20–0.73, 1383.12–2359.91 kg/hm$^2$, and 1.14–1.94. Model 15 based on CIred edge proved to have good fitting characteristics, relatively high precision models.

At the pod stage, $R^2_{cal}$ and RMSEC of calibration models varied, respectively, from 0.22–0.68 and 2570.50–4003.84 kg/hm$^2$, while the $R^2_{val}$, RMSEV, and RPD of the validation models were between 0.20–0.61, 2768.50–3959.06 kg/hm$^2$, and 1.10–1.57; models 24, 25, and 26 based on NDVI, GNDVI, and RVI proved to have good fitting characteristics, relatively high precision models, respectively.

Figure 4 showed the relationship between CIred edge and AGB. CIred edge performed better in estimating AGB among the seven VIs at four different growth stages. However, the results indicated a low correlation between CIred edge and AGB at the flowering and pod stage. During reproductive growth (flowering and pod stage), photosynthetic products were mostly stored in reproductive organs (e.g., flower and pod), while photosynthetic products were mostly stored in stems and leaves during vegetative growth [40]. The canopy morphology structure of WOR also changed significantly. Thus, it was difficult to monitor AGB by VIs that are based on visible and near infrared bands.

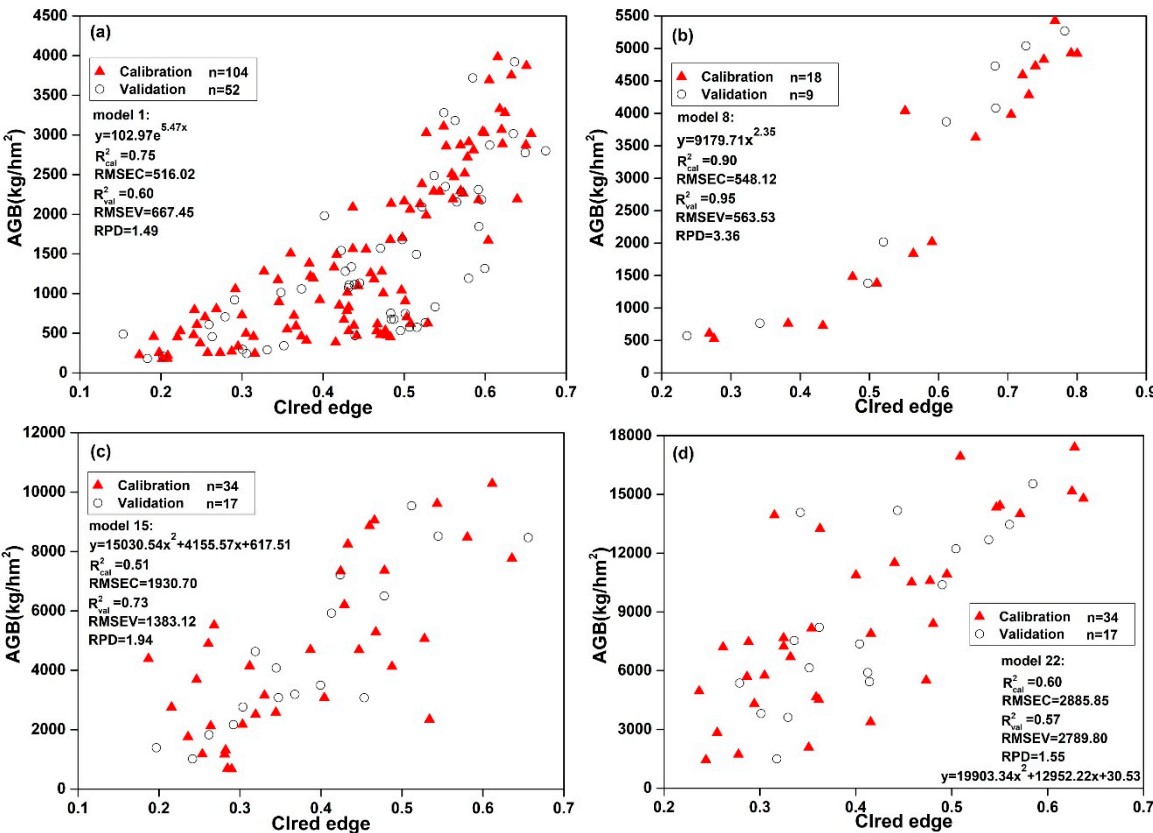

**Figure 4.** Relationship between CIred edge and AGB at four different stages: (**a**) seeding stage; (**b**) bolting stage; (**c**) flowering stage; and, (**d**) pod stage.

### 3.5. The Coefficient of Determination ($R^2$) Between AGB and Narrow Band Indices

In this study, the narrow band NDVI-like indices were calculated in the form of NDVI using all possible two-band combinations; the following equations:

$$\text{NDVI-like} = \frac{R_i - R_j}{R_i + R_j} \tag{9}$$

Where $R_i$ and $R_j$ are the narrow band reflectance at wavelength i and j, with i < j, respectively.

Through the linear regression analysis of AGB and NDVI-like indices, the determination coefficient ($R^2$) values were calculated using the calibration datasets (Figure 5). The dashed rectangular boxes indicated that NDVI-like indices ($R_i$: 750–1150 nm, $R_j$: 1150–1300 nm) proved to have relatively higher and stable $R^2$ values at four different growth stages, and $R^2$ values ranging from 0.45 to 0.55 at the seeding stage, 0.47 to 0.65 at the bolting stage, 0.58 to 0.72 at the flowering stage, and 0.47 to 0.74 at the pod stage, respectively. However, the $R^2$ values of NDVI-like indices, which combined visible bands (400–700 nm) and NIR bands (750–1300 nm), were relatively low.

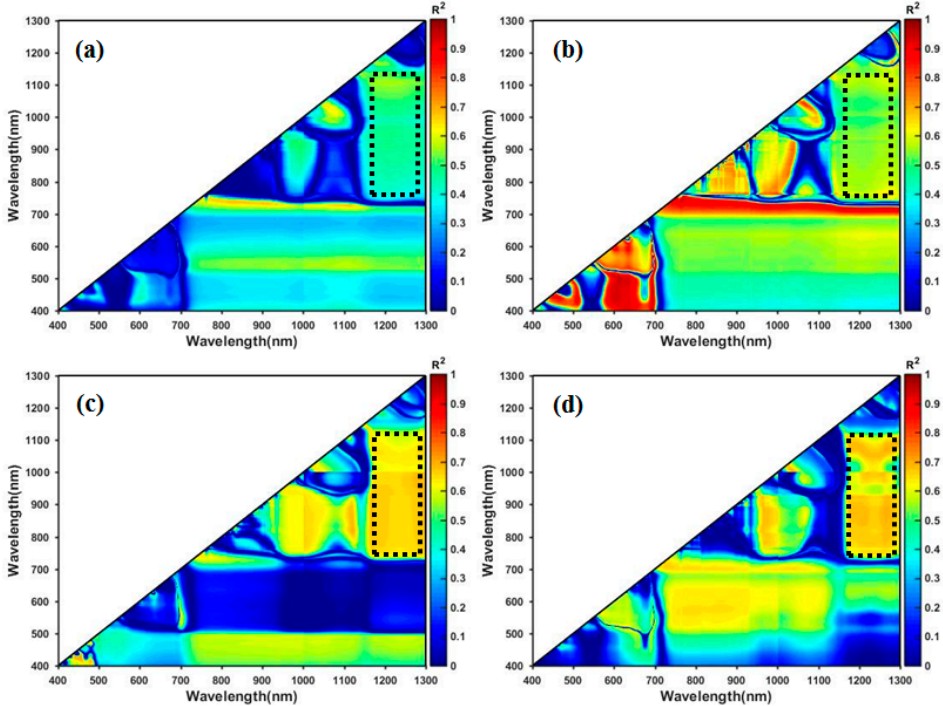

**Figure 5.** The coefficient of determination ($R^2$) between AGB and NDVI-like indices at four different stages: (**a**) seeding stage; (**b**) bolting stage; (**c**) flowering stage; and, (**d**) pod stage.

### 3.6. Extract the Feature Wavelengths by PLSR

To further extract the feature wavelengths for improving AGB estimation accuracy, the variable importance in projection (VIP) scores from PLSR was applied to evaluate the performance of the feature bands with the combination of four different grown stages data. The VIP could indicate the contribution of each wavelength (x, independent variable) to the value of AGB (y, dependent variable); it is calculated by equations (10).

$$\text{VIP}_k(a) = K\sum_a W^2 ak\left(\frac{SSY_a}{SSY_t}\right) \tag{10}$$

Where $\text{VIP}_k(a)$ is the importance of the wavelength variable based on a model with a factors (PLSR-components); Wak is the PLSR-weights of wavelength variable in an ath PLSR-factor; $SSY_a$ represents the explained sum of squares of Y; and, $SSY_t$ is the total sum of squares of Y. The threshold of the VIP score is 1.0 and a large VIP value shows that the wavelength is more important for AGB estimation.

Figure 6 showed that the VIP scores of visible wavelengths (400–750 nm) were all below 1.0 (the threshold of the VIP score is 1.0), and large VIP values are located at 800 nm and 1200 nm; it indicated that the NIR wavelengths were more important than the visible wavelengths for WOR AGB estimation. It was possible to construct new vegetation indices only from NIR wavelengths to improve WOR AGB estimation accuracy and the wavelengths (800 and 1200 nm) could be selected as the feature wavelengths.

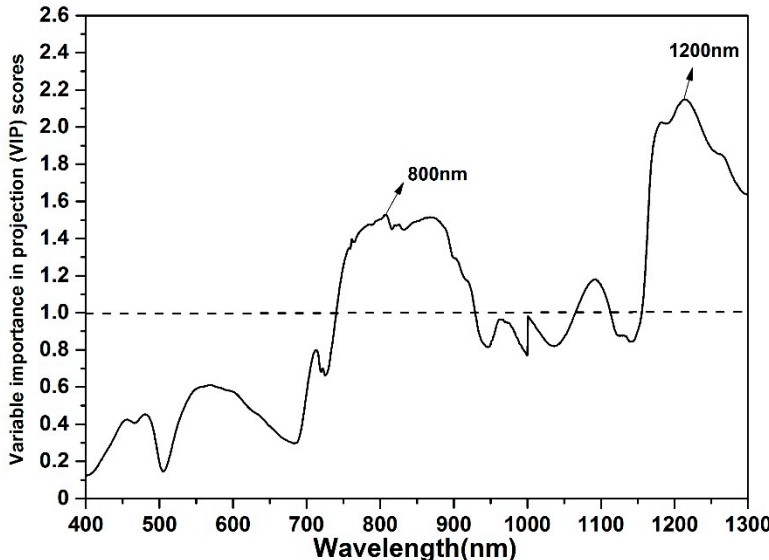

**Figure 6.** The partial least squares regression variable importance in projection (PLSR VIP) of each wavelength with the combination of four different grown stages data

### 3.7. Relationship Between NDVI (800, 1200) and AGB

We used linear and nonlinear regression analysis to investigate NDVI (800, 1200) with AGB. Table 6 indicated that the nonlinear regression models that are based on NDVI (800, 1200) performed well at four different growth stages; NDVI (800, 1200) had a good correlation with the AGB of WOR. The determination coefficients $R^2_{cal}$ and $R^2_{val}$ were greater than 0.60, the values of RMSEC and RMSEV varied from 622.82 to 2944.28 kg/hm$^2$, and the ratio of prediction to deviation (RPD) are distributed between 1.45 and 1.99. Most red and near-infrared-based spectral VIs (NDVI, SAVI) had a low correlation with AGB at the flowering and pod stages. However, NDVI (800, 1200) was highly correlated with AGB at the flowering and pod stages, and the two effective wavelengths (800 nm and 1200 nm) proved to have good fitting characteristics in estimating AGB.

**Table 6.** Regression modeling analysis between NDVI (800, 1200) and AGB.

| Growth Period | Model Number | Vegetation Index | Model | $R^2_{cal}$ | RMSEC | $R^2_{val}$ | RMSEV | RPD |
|---|---|---|---|---|---|---|---|---|
| Seeding stage | 29 | NDVI (800, 1200) | $Y = 136.47e^{19.92X}$ | 0.64 | 622.82 | 0.61 | 650.03 | 1.45 |
| Bolting stage | 30 | NDVI (800, 1200) | $Y = 293314.87X^2 - 38660.43X + 1223.73$ | 0.70 | 964.67 | 0.80 | 963.94 | 1.96 |
| Flowering stage | 31 | NDVI (800, 1200) | $Y = 83018.78X^2 + 3179.73X + 867.19$ | 0.67 | 1588.22 | 0.74 | 1351.36 | 1.99 |
| Pod stage | 32 | NDVI (800, 1200) | $Y = 464172.09X^2 - 115386.42X + 8971.06$ | 0.77 | 2200.45 | 0.62 | 2944.28 | 1.47 |

### 3.8. Regression Modeling Analysis for AGB of Different Growth Stages Combination

In this study, in combining WOR growth period, the growth stages of WOR could be divided into two stages: the vegetative stage (including seeding and bolting stages) and reproductive stage (including flowering and pod stages). The AGB estimation of different combined growth stages could test the robustness of NDVI (800, 1200) and the CIred edge.

The results showed that the CIred edge was effective in estimating AGB at the vegetative stage, with the values of $R^2_{cal}$, $R^2_{val}$, RMSEC, RMSEV, and RPD being 0.84, 0.75, 525.35 kg/hm$^2$, 643.48 kg/hm$^2$, and 2.01. However, the inversion of the CIred edge and AGB showed a low coefficient of determination at the reproductive stage and the total for four stages, with the value of $R^2_{cal}$, $R^2_{val}$, RMSEC, RMSEV, and RPD being 0.08, 0.06, 3507.79 kg/hm$^2$, 3447.41 kg/hm$^2$, and 1.03 at the total for four stages.

NDVI (800, 1200) had good robustness at the vegetative and reproductive stage, and it significantly improved the AGB estimation accuracy at the total for four stages; $R^2_{cal}$ ranged from 0.71 to 0.83, $R^2_{val}$ ranged from 0.52 to 0.82, RMSEC ranged from 702.88 to 2226.83 kg/hm$^2$, RMSEV ranged from

889.44 to 2232.90 kg/hm$^2$, and RPD ranged from 1.45 to 2.37. Through comparing the accuracy of models, NDVI (800, 1200) proved to have good fitting characteristics for the estimation of AGB at different stages.

The VIs-AGB scatter plots showed the distribution of calibration and validation dataset. The result of model 35 indicated that NDVI (800, 1200) was more effective and accurate than CIred edge in AGB estimation at the reproductive stage, and the distribution of scatter points was also more regular. When data from all four growing stages were used in the modeling, AGB estimation precision of model 37 increased significantly, with the value of $R^2_{cal}$, $R^2_{val}$, RMSEC, RMSEV, and RPD were 0.83, 0.82, 1521.84 kg/hm$^2$, 1501.34 kg/hm$^2$, and 2.37. However, the distribution scatter points of model 38 were clearly divided into two parts with a poor fitting performance. The above results indicated that NDVI (800, 1200) could be used to obtain stable AGB estimation models.

## 4. Discussion

### 4.1. Estimating WOR AGB Using the Seven Commonly Used VIs

In this study, different nitrogen rates in WOR experiments were conducted in Wuxue city and Shayang city, Hubei province, central China from 2014 to 2016 to create different biomass condition. Canopy hyperspectral data and AGB samples were collected at four growth stages from seeding through pod stage. The linear and non-linear statistical models that simulated the quantitative relation between the seven commonly used VIs (CIred edge, CIgreen, NDVI, GNDVI, RVI, DVI, and SAVI) and AGB at different growth stages.

Through comparing the accuracy of models (Table 5), we could see that the growth stage had a strong influence on the sensitivity to different wavelengths and performance of VIs for estimating AGB [49,50]. The non-linear models based on the CIred edge showed excellent performance, with higher accuracy ($R^2_{cal}$ ranged from 0.75–0.90, $R^2_{val}$ ranged from 0.60–0.95, respectively) when compared to the other six VIs at the seeding and bolting stage. However, CIred edge performed significantly worse ($R^2_{cal}$ ranged from 0.51–0.60, $R^2_{val}$ ranged from 0.57–0.73, respectively) at the flowering and pod stage. The CIred edge was effective in estimating AGB at the vegetative stage, while it had relatively lower precision in the reproductive stage (Table 7). Additionally, the scatterplots also showed that, in the reproductive stage, the samples were distributed away from the regression curve (Figure 7d).

**Table 7.** Regression modeling analysis between NDVI (800, 1200), CIred edge, and AGB.

| Growth Period | Model Number | Vegetation Index | Model | $R_{cal}^2$ | RMSEC | $R_{val}^2$ | RMSEV | RPD |
|---|---|---|---|---|---|---|---|---|
| Vegetative stage | 33 | NDVI (800, 1200) | $Y = 165728.69X^2 - 10401.76X + 391.91$ | 0.71 | 702.88 | 0.52 | 889.44 | 1.45 |
| | 34 | CIred edge | $Y = 15403.76X^2 - 6617.28X + 1101.59$ | 0.84 | 525.35 | 0.75 | 643.48 | 2.01 |
| Reproductive stage | 35 | NDVI (800, 1200) | $Y = 467.84e^{11.51X}$ | 0.73 | 2226.83 | 0.72 | 2231.90 | 1.85 |
| | 36 | CIred edge | $Y = 20042.96X^2 + 7467.16X + 377.43$ | 0.46 | 3159.01 | 0.48 | 2938.05 | 1.41 |
| Total for four stages | 37 | NDVI (800, 1200) | $Y = 197455.98X^2 - 17815.49X + 779.08$ | 0.83 | 1521.84 | 0.82 | 1501.34 | 2.37 |
| | 38 | CIrededge | $Y = -5472.86X^2 + 12188.17X - 717.08$ | 0.08 | 3507.79 | 0.06 | 3447.41 | 1.03 |

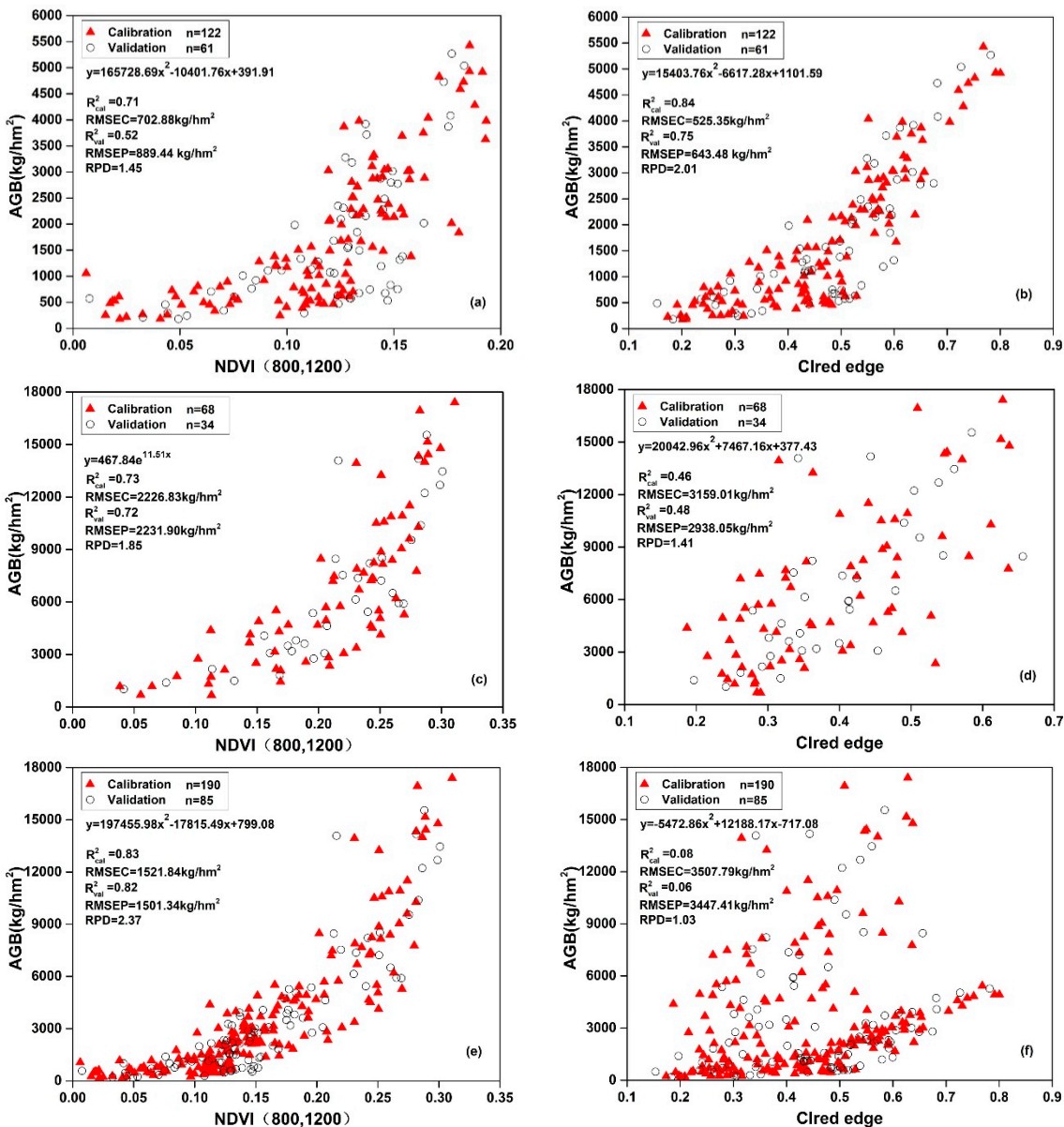

**Figure 7.** Regression modeling analysis between NDVI (800, 1200), CIred edge, and AGB at different stages: (**a**) model 33; (**b**) model 34; (**c**) model 35; (**d**) model 36; (**e**) model 37; and, (**f**) model 38.

Leaves dominated WOR AGB in the vegetative stage, while the stem, flowers, and pod biomass contributed the most of AGB at reproductive stage [39,51]. As Figure 3 showed that flowers affected the canopy spectral reflectance, with visible bands (500–730 nm) increasing significantly and NIR bands (800–1300 nm) decreasing slightly. CIred edge is calculated by the red-edge band (725 nm) and NIR band (800 nm), which relies primarily on the sensitivity of red edge bands to leaf chlorophyll absorption [52], the red-edge region could provide a possible way to avoid the spectral saturation for vegetation with moderate to high density [29]. Therefore, the CIred edge could effectively estimate AGB when WOR were in a period of vigorous green leaves [53], but it was not suitable for estimating AGB at the reproductive stage.

### 4.2. Estimating WOR AGB Using the Modified Vegetation indices NDVI (800, 1200)

The NDVI-like indices and the PLSR VIP of each wavelength were calculated to further extract the feature wavelengths. The $R^2$ values were also calculated between NDVI-like indices and AGB at four different growth stages. Figure 5 indicated that NDVI-like indices that were based on NIR bands

($R_i$: 750–1150 nm, $R_j$: 1150–1300 nm) proved to have a relatively good correlation with the AGB. The $R^2$ values gradually increased along the development of WOR phenology, and ranged from 0.45 to 0.55 at the seeding stage, 0.47 to 0.65 at the bolting stage, 0.58 to 0.72 at the flowering stage, and 0.47 to 0.74 at the pod stage, respectively. These results demonstrated that NDVI-like indices that were calculated by simple combinations of NIR wavelengths were more sensitive to AGB, which was consistent with previous results [54].

The two effective wavelengths (800 nm and 1200 nm) that were extracted by the PLSR VIP had clear physical significance and higher recognition; the effective wavelength of 800 nm was usually used to construct many commonly used vegetation indices for estimating crop growth parameters, such as leaf area index [55], biomass [56], leaf nitrogen content [34], and canopy chlorophyll content [57]; the canopy reflectance of 800 nm was not only relatively stable, but it was also insensitive for leaf structure. Besides, the wavelength around 1200 nm proved to be sensitive for plant moisture [58] and AGB [6].

As Table 6 showed that NDVI (800, 1200) that was based on two effective wavelengths (800 nm and 1200 nm) could have considerably good fitting characteristics in estimating WOR AGB at different growth stages, especially at the flowering and pod stages. When comparing with the CIred edge (Table 7), NDVI (800, 1200) also retained good robustness at the vegetative stage and reproductive stage, and it could reduce the influence of flower and pod for estimating AGB. Furthermore, model 37, based on NDVI (800, 1200), significantly improved the AGB estimation accuracy ($R^2 > 0.80$, RMSE < 1530 kg/hm$^2$, RPD > 2.3) without saturation phenomenon at total for four stages. The results showed that the modified vegetation indices NDVI (800, 1200) could be used to estimate WOR AGB at different growth stages.

## 5. Conclusions

In this study, we conducted the WOR AGB estimates using vegetation indices (VIs) that are derived from canopy hyperspectral data at different growth stages (seeding stage, bolting stage, flowering stage, and pod stage). The different fertilizer-N gradients field experiments were conducted in two consecutive WOR growing season spanning from September 2014 to May 2015 in Wuxue city and September 2015 to May 2016 Shayang city. Correlation analyses were performed between canopy hyperspectral data and AGB. The linear and non-linear (logarithmic, parabolic, power, and exponential) regression statistical models simulated the quantitative relation between VIs and AGB at different growth stages.

Through comprehensive analysis of the AGB estimation models, the results demonstrated that WOR AGB could be accurately estimated by canopy hyperspectral data at different growth stages. CIred edge had relatively higher precision for AGB estimation at the vegetative stage, but it was not effective in estimating AGB at the reproductive stage, because of the disturbance of flowers and pods on visible bands. NDVI (800, 1200) retained good robustness at the vegetative stage and reproductive stage, it could reduce the influence of flower and pod in estimating AGB, the model 37 based on NDVI (800, 1200) significantly improved AGB estimation accuracy ($R^2 > 0.80$, RMSE < 1530 kg/hm$^2$, RPD > 2.3) without saturation phenomenon at total for four stages. We concluded that NDVI (800, 1200) could be used to accurately estimate WOR AGB at different growth stages. Moreover, it was vital to pay more attention to near-infrared (NIR) bands that could represent the phenology of vegetation growth [33]. Selecting suitable VIs and modeling algorithms could significantly affect the success of AGB estimation of the crops with variable plant morphology [59], such as winter oilseed rape.

Although the appropriate VIs and effective wavelengths that are described in this paper were constructed using the data collected in different years, growth stages, and fertilizer-N level; it is worth noting that this newly developed estimation based on empirical models might be site-specific, and more field studies are needed to examine their robustness of other oilseed rape cultivars under a wider range of conditions. Future work will focus on exploring new vegetation indices and wavelengths for improving the accurate and robust estimation of large-scale WOR AGB, and combining different platforms that carried various remote sensing sensors for precision agriculture, including ground

platform, unmanned erial vehicle (UAV) platform, and satellite platform; the application of multiscale remote sensing technology will help farmers to monitor crop growth status accurately and properly control fertilizing amount.

**Author Contributions:** All authors have made significant contributions to this research. S.F. designed the paper and experiments, Y.M. and Y.P. carried out the analysis and provided the writing of this paper. Y.G. and D.W. provided important insights and suggestions on this research. All authors read and approved the final manuscript.

**Funding:** This research was supported by the National 863 Project of China (2013AA102401).

**Acknowledgments:** We acknowledge the support and the use of the facilities and equipment provided by the School of Remote Sensing and Information Engineering, Wuhan University, China. We are very thankful to the research groups led by Shanqin Wang and Jianwei Lu, College of Resources and Environment, Huazhong Agricultural University, China for your hard work to design experiment and collect field samples.

**Conflicts of Interest:** The authors declare no conflicts of interest.

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
