# Peer review of "Remote Estimation of Biomass in Winter Oilseed Rape (Brassica napus L.) Using Canopy Hyperspectral Data at Different Growth Stages"

_applsci, doi:10.3390/app9030545_

Round 1

Reviewer 1 Report

The paper is very interesting but unoriginal, however it is well written. Also, it is appropriate for this journal. One questions:

In the models linear, the standardized coefficients (also called beta coefficients) are used to compare the relative wights of the variables. The higher the absolute value of a coefficient, the more important the wight of the corresponding variable. When the confidence interval around standardized  coefficients has value 0 (this can be easily seen on the chart of normalized coefficients), the weight of a variable in the model is not significant. The table of standardized coefficients (also called beta coefficients) is necessary, also graphics.

Although it is a minor revision it may take some time.

Author Response

Dear  reviewers:

Point 1: In the models linear, the standardized coefficients (also called beta coefficients) are used to compare the relative weights of the variables. The higher the absolute value of a coefficient, the more important the weight of the corresponding variable. When the confidence interval around standardized coefficients has value 0 (this can be easily seen on the chart of normalized coefficients), the weight of a variable in the model is not significant. The table of standardized coefficients (also called beta coefficients) is necessary, also graphics.

Although it is a minor revision it may take some time.

 Response 1: Thank you for your time and efforts to review our manuscript. We sincerely appreciate for your valuable comments, but the purpose of our manuscript is to extract the effective wavelengths and construct new vegetation indices to reduce the influence of flower and pod, and select simple regression model forms (linear, logarithmic, parabolic, power and exponential) which are suitable for accurately remote estimation of winter oilseed rape biomass at different growth stages.

As far as we know, the standardized coefficients (also called beta coefficients) are mainly applied to select variables in the multiple linear regressions, our manuscript are not related to the multiple linear regressions in the data processing. We extracted the feature wavelengths by correlation analysis (Figure 5, the line 272-292-revision) and the VIP of PLSR (Figure 6, the line 293-312-revision) for constructing new vegetation indices. The specific data processing refers to the line (163-194-revision), the linear and non-linear regression models were established based on new vegetation indices and biomass of the calibration dataset, we selected the best model forms (linear, logarithmic, parabolic, power and exponential) by comparing the R2 and RMSEC of calibration models, and the R2, RMSEV and RPD were used to evaluate the accuracy of validation models. Although we do not calculate the standardized coefficients to compare the relative weights of the variables, your suggestion gives us some inspiration to synthesize multiple vegetation indices for the multiple linear regressions, and select more suitable combination of vegetation indices for large-scale winter oilseed rape biomass estimation in the follow-up study.

Reviewer 2 Report

Introduction - This should give more context into the benefits of remote sensing - targeting N fertiliser applications, sustainability benefits etc. Good amount of literature cited for predictions of biomass from remote sensing.

136 - When the biomass is calculated using the unified density, this assumes all transplanted plants survive. Is this correct, surely if there are some reductions in plant density this will have an impact on the biomass value.

137 - This needs further clarification to help the reader understand what plots/samples are made to get the numbers in Table 2.

Figure 4 label - Should say validation not valibration

Figure 4 - needs more clarification about validation process. Presumably the equations derived from the calibration data were used to predict the AGB for the different indices.

304 - Would the 'bolting' stage normally be considered as vegetative growth?

404 - What about variety? There is potential for your estimations may be variety specific.

Conclusions - what is the benefit of the research to farmers?

The English of the paper could be improved - a number of spelling and grammatical errors throughout.

Author Response

Dear reviewers:

Thank you for your time and efforts to help us improve our manuscript. We sincerely appreciate for the valuable suggestions and comments, which indeed help us a lot.

Point 1: Introduction - This should give more context into the benefits of remote sensing - targeting N fertiliser applications, sustainability benefits etc. Good amount of literature cited for predictions of biomass from remote sensing.

Response 1: Thank you for this comment. Remote sensing technology has indeed been applied in many aspects of precision agriculture. We have added some recent references to the application of remote sensing for monitoring oilseed rape nitrogen and chlorophyll (the line 41-49-revision). The aim of our manuscript was to remote estimation of biomass in winter oilseed rape, so we mainly searched and quoted some literatures about the predictions of biomass from remote sensing, especially the recent literatures about oilseed rape biomass.

Point 2: 136 -When the biomass is calculated using the unified density, this assumes all transplanted plants survive. Is this correct, surely if there are some reductions in plant density this will have an impact on the biomass value.

Response 2: In the course of two years experiments, all the transplanted oilseed rape survived under normal management decisions (pest control and herbicide application) followed the local standard practices without the emergence of extreme weather. As far as we know, plant density has a certain effect on biomass value, relevant literatures[1-3] indicate that oilseed rape biomass will increase with the raising of plant density in the normal plant density range, the effect of plant density on remote estimation of oilseed rape biomass needs further study.

References:

1.  Li, M.; Naeem, M.S.; Ali, S.; Zhang, L.; Liu, L.; Ma, N.; Zhang, C. Leaf Senescence, Root Morphology, and Seed Yield of Winter Oilseed Rape (Brassica napus L.) at Varying Plant Densities. Biomed Research International 2017, 2017, 8581072. (page 11 , table.3)

2.  Ejj, M.; Zhou, W. Growth and yield responses to plant density and stage of transplanting in winter oilseed rape (Brassica napus L.). Journal of Agronomy and Crop Science 2010, 186, 253-259. (page 6 , table. 4)

3.  Wang, R.; Cheng, T.; Hu, L. Effect of wide-narrow row arrangement and plant density on yield and radiation use efficiency of mechanized direct-seeded canola in Central China. Field Crops Research 2015, 172, 42-52, doi:10.1016/j.fcr.2014.12.005. (page 6 , Fig. 4)

Point 3: 137-this needs further clarification to help the reader understand what plots/samples is made to get the numbers in Table 2.

Response 3: As suggested, we added some information to help the reader understand the source of samples, such as “Study Site”, “Sampling dates” and “Number of samples” (the line 145-152-revision). Sampled data at the each time was divided into 2:1 ratio as calibration dataset (the first two replications) and validation dataset (the third replication) corresponding the plots which were showed in the Figure 1 (the line 105-107-revision).

Point 4: Figure 4 label - Should say validation not valibration

Response 4: Thank you for reminding us of the spelling mistake. We have checked and corrected some spelling and grammatical errors in our manuscript (Figure 4 label, the line 268-269-revision) and (Figure 7 label, the line 342-345-revision).

Point 5: Figure 4 - needs more clarification about validation process. Presumably the equations derived from the calibration data were used to predict the AGB for the different indices.

Response 5: We try to add some detailed description about “Data modeling process” (the line 178-185-revision). The predicted AGB value was calculated by vegetation indices (VIs) of the validation dataset with the application of the selected best calibration models forms and its corresponding parameters. In other words, the best equations derived from VIs and AGB of the calibration dataset were only used to predict the AGB by the same VIs of the validation dataset, not by other VIs.

Point 6: 304-Would the 'bolting' stage normally be considered as vegetative growth?

Response 6: The bolting stage of oilseed rape is a transitional stage from vegetative growth to reproductive growth; relevant literature indicates [4,5] that vegetative growth is the most vigorous at bolting stage, nutritional growth is dominant, there are three kinds of oilseed rape leaves: long petiole, long petiole and sessile leaves at bolting stage[6]. Meanwhile, there are no obvious reproductive growth labels such as flowers and pods at sampling dates of our bolting stage. Therefore, it is reasonable to incorporate bolting stage into vegetative growth stage in this article.

References:

4. Zhong, F; Zhang, M; Dai, R; JIANG, H; Zhou, Q. Alleviative effects of triadimefon( TDM) on the growth and antioxidant in oilrape under drought at the bolting-stage. Journal of Nanjing Agricultural University. 2016, 39, 730-738.

5. Tao, x; Li, h; Wan, L; Zhou, Q; JIANG, H. Alleviation effects of IAA foliar spray on waterlogging stressed rapeseed at budding stage. Chinese Journal of Oil Crop Sciences. 2015, 37, 55-61.

6.  LU, Z; REN, T; LU, J; LI, X; CONG, R; PAN, Y; LI, K. Main factors and mechanism leading to the decrease of photosynthetic efficiency of oilseed rape exposure to potassium deficiency. Journal of Plant Nutrition and Fertilizer. 2016, 22, 122-131.

Point 7: 404-What about variety? There is potential for your estimations may be variety specific.

Response 7: Although the oilseed rape variety selected in this article was a widely planted variety in the local experiment area (the line 123-revision), our results still had some limitations, more field studies are needed to examine empirical models robustness of other oilseed rape cultivars under a wider range of conditions in the future(the line 436-445-revision).

Point 8: Conclusions - what is the benefit of the research to farmers?

Response 8: The dry aboveground biomass (AGB) is an important parameter to indicate winter oilseed rape growth status; farmers need biomass information at different growth stages for guiding their applying fertilizer and yield prediction. Remote sensing provided a cost-effective and quantitative method for avoiding sampling bias at both local and regional scale estimation of oilseed rape AGB. As oilseed rape is very different from rice, wheat, and soybean, as it is a broadleaf plant with an obvious distinctive change of canopy structure during its entire growing period, and the remote sensing data is also seriously affected by its yellow flowers and green pods. Through the analysis of our manuscript, remote sensing could more accurately estimate oilseed rape AGB at different growth stages. If we used the extract the effective wavelengths and construct new vegetation indices from our manuscript by combining various remote sensing sensors, it is possible for us to achieve the large-scale oilseed rape AGB estimation accurately, and help farmers to monitor crop growth status accurately and properly control fertilizing amount. We added some prospects for future research (conclusions , the line 437-446-revision).

Point 9: The English of the paper could be improved - a number of spelling and grammatical errors throughout.

Response 9: Thank you sincerely for the valuable comments, we have checked and corrected some spelling and grammatical errors in full text.
